# Ultrasound Assisted Coextraction of Cornicabra Olives and Thyme to Obtain Flavored Olive Oils

**DOI:** 10.3390/molecules28196898

**Published:** 2023-10-01

**Authors:** Fátima Peres, Madalena Pinho Marques, Miguel Mourato, Luisa L. Martins, Suzana Ferreira-Dias

**Affiliations:** 1Instituto Politécnico de Castelo Branco, Escola Superior Agrária, 6000-909 Castelo Branco, Portugal; fperes@ipcb.pt; 2LEAF—Linking Landscape, Environment, Agriculture and Food Research Center, Associated Laboratory TERRA, Instituto Superior de Agronomia, Universidade de Lisboa, Tapada da Ajuda, 1349-017 Lisboa, Portugal; madalena.marques@live.com (M.P.M.); mmourato@isa.ulisboa.pt (M.M.); luisalouro@isa.ulisboa.pt (L.L.M.); 3Laboratório de Estudos Técnicos, Instituto Superior de Agronomia, Universidade de Lisboa, Tapada da Ajuda, 1349-017 Lisboa, Portugal

**Keywords:** enriched olive oils, lemon thyme, mastic thyme, phenolic profile, oxidative stability

## Abstract

Flavoring olive oils is a new trend in consumer preferences, and different enrichment techniques can be used. Coextraction of olives with a flavoring agent is an option for obtaining a flavored product without the need for further operations. Moreover, ultrasound (US) assisted extraction is an emergent technology able to increase extractability. Combining US and coextraction, it is possible to obtain new products using different types of olives (e.g., cultivar and ripening stage), ingredient(s) with the greatest flavoring and/or bioactive potential, as well as extraction conditions. In the present study, mastic thyme (*Thymus mastichina* L.) (TM) and lemon thyme (*Thymus x citriodorus*) (TC) were used for flavoring Cornicabra oils by coextraction. The coextraction trials were performed by (i) thyme addition to the olives during crushing or malaxation and (ii) US application before malaxation. Several parameters were evaluated in the oil: quality criteria parameters, total phenols, fatty acid composition, chlorophyll pigments, phenolic profile and oxidative stability. US application did not change the phenolic profile of Cornicabra olive oils, while the enrichment of olive oils with phenolic compounds or pigments by coextraction was very dependent on the thyme used. TM enrichment showed an improvement of several new phenolic compounds in the oils, while with TC, fewer new phenols were observed. In turn, in the trials with TC, the extraction of chlorophyll pigments was higher, particularly in crushing coprocessing. Moreover, the oils obtained with US and TM added in the mill or in the malaxator showed lower phenol decrease (59%) than oils flavored with TC (76% decrease) or Cornicabra virgin olive oil (80% decrease) over an 8-month storage period. Multivariate data analysis, considering quality parameters, pigments and phenolic contents, showed that flavored oils were mainly grouped by age.

## 1. Introduction

Olive oil is the main lipid source of the Mediterranean diet, and more than 85% of world production is concentrated in this region [1]. Moreover, spices and herbs are traditionally added to olive oil in Mediterranean gastronomy to enhance its flavor, improving food pairing experiences [2]. Therefore, developing new products combining olive oil and herbs can be a strategy for obtaining—in a “ready to use” manner—a product, which enhances the properties of each component. Moreover, these products are innovative foods identified with the Mediterranean diet.

Virgin olive oil (VOO) extraction mainly involves four steps: crushing, malaxation, solid/liquid separation and liquid/liquid separation. Therefore, only mechanical means are allowed to be used in the extraction process. The quality of the fruits is the main factor influencing the final oil quality. Therefore, when overripe fruits are used, some flavor characteristics of the olives may be lost.

Several enrichment techniques can be used for flavoring olive oils: addition of aromas/bioactive extracts/essential oils [3,4], infusion [5] or coextraction [6,7]. This last technique is based on the addition of herbs or other vegetable materials in the crushing or malaxation steps [6,8]. However, according to European legislation, a product obtained by coextraction cannot be labeled as “virgin olive oil”. In this case, as the flavoring process is carried out during milling or beating the olives, the term “olive oil” cannot be used on the label. Other label designations, such as “food preparation based on olives and thyme” or “food compound of olives and thyme” or “culinary product made from olives and thyme”, should be used [9].

Recently, emerging technologies, such as high pressure processing (HPP) [10], pulsed electric fields (PEF) [11,12] or ultrasound assisted extraction (USAE) [13,14], are being applied in olive oil technology to increase environmental sustainability by improving VOO extraction yields [15,16,17]. In the case of USAE, two mechanisms could be useful for optimizing the VOO extraction process: the mechanical and the thermal effects [16]. Mechanical action is due to the cavitation phenomena, which enables accelerated heat and mass transfer [14]. In fact, ultrasound frequency (20–100 kHz) involves the mechanical primary mechanism of action of acoustic cavitation, which is the formation, growth and final collapse of microbubbles within a surrounding medium due to pressure fluctuations induced by the applied sound field [18]. The use of ultrasound (US) in olive oil extraction has, as the main objective, the rupture of the oil-bearing cells not yet destroyed by crushing. This will lead to the release of oil, increasing the extraction yield. Moreover, the thermal effect of US will allow the reduction in malaxation time [16]. In addition, the use of US showed a limited modification of phenolic and volatile composition of the olive oils [19]. Consequently, USAE can be useful for promoting the aromatization process of the olive oil, facilitating the extraction and the dissolution of bioactive compounds into the oil [17,20,21].

The present study consists of coextraction of “Cornicabra” cultivar olives with lemon thyme (*Thymus x citriodorus*; TC) or mastic thyme (*Thymus mastichina* L.; TM) to obtain new olive oil preparations. Thyme addition was performed either during crushing or malaxation. The effect of USAE before malaxation was also evaluated. The effects of the coextraction technique used and the conditions of USAE on quality parameters, phenolic composition and oxidative stability of the obtained products over an 8-month storage period were evaluated. As far as we know, this is the first study on the coextraction of Cornicabra olives with two different types of thymes and with USAE application. 

## 2. Results 

### 2.1. USAE Trials

After “Cornicabra” olive milling, the paste was submitted to ultrasound assisted extraction with a frequency of 35 kHz and 10 min US application time, followed by a 20 min malaxation, which resulted in a total treatment time of 30 min. The phenolic content of this olive oil was 325.15 ± 4.90 mg GAE/kg oil, while the phenolic content of olive oil obtained without US application (30 min malaxation) was 264.22 ± 3.99 mg GAE/kg oil. The use of US promoted a significant increase of about 23% in olive oil phenolic content. 

Therefore, an US pretreatment for 10 min at a frequency of 35 kHz, followed by a 20 min malaxation, was used in the subsequent coextraction experiments with both thymes.

### 2.2. Characterization of Oil Preparations

Olive oil flavored preparations were characterized by several parameters specific to olive oil, according to European regulations (quality criteria, main fatty acid composition), as well as total phenols and phenolic profile after extraction and at two storage time points (four and eight months). The oxidative stability of oil preparations is also discussed.

#### Quality Criteria

The main problem with the addition of herbs to olive oils is the incorporation of new products in the oil, which could promote the main degradation reactions, such as lipolysis and oxidation. The risk of lipolysis increases when fresh herbs are used, as the presence of free water in the oil will increase triacylglycerol’s hydrolytic reactions [22,23,24]. Regarding the oxidation reactions, the fatty acid composition of oils is a determining factor for oxidation susceptibility [25].

Table 1 presents the results of quality criteria for Cornicabra oils obtained in an Abencor system either with or without USAE treatment. As can be observed, these oils, although extracted from olives with a high ripening index (late harvest), have very low acidity due to the low degradation of the fruits. In fact, the Beira Alta Region, where the fruits were collected, has a climate, which enables pests and diseases of the olive tree to be controlled through low temperatures during olive ripening. The peroxide value (PV) is also far from the limit for virgin olive oils (20 meqO_2_/kg), as is the specific absorbance at 270 nm, K_270_ (legal limit: ≤0.22). Specific absorbances at 232 nm, K_232_ (legal limit: ≤2.50) are related to the high contents of polyunsaturated fatty acids (PUFA) of this cultivar. The rates of autoxidation of these fatty acids are faster, and linoleic acid hydroperoxide and the conjugated dienes, which may result from its decomposition, show an absorption band at 232 nm. Regarding organoleptic assessment, no defects were detected, and a ripe olive fruit intensity around 2.0 was detected by the sensory panel, along with a sensation of ripe fruits, such as peach. Therefore, the quality criteria indicate that both Cornicabra oils obtained with or without USAE are classified as extra virgin olive oils.

Table 2 presents the results of quality criteria for flavored oil preparations with lemon thyme (TC) or mastic thyme (TM) addition in the crushing step (m: milling) or before malaxation operation (b: beating). As can be seen, the chemical quality parameters are within the limits for extra virgin olive oil, and no significant differences are observed for each parameter among the various oil preparations obtained with TC or TM, added either in the milling or malaxation steps, except for TCm, which shows a slight increase in Peroxide Value.

With respect to the sensory analysis, none of the flavored preparations presented defects. The oils obtained in the absence of ultrasound treatment had a thyme flavor around 3.5–4.9. The thyme flavor was more intense when lemon thyme (TC) was added in the crusher (TCb) than in the malaxator (TCm) (3.5 vs. 3.0, respectively). The US treatment promoted a decrease in thyme flavor in TC coextracted oils (intensity of 1.5 and 2.5 in usTCb and usTCm, respectively). However, for mastic thyme, the thyme intensity was similar, both in preparations where it was added in the mill (TMm) and in the malaxator (TMb). Flavored oil preparations, as well as the virgin olive oil (Table 1 and Table 2), presented low bitter intensity, and their intensity seemed not to be affected by US treatment. Pungency varied between 1.2 (usTMm) and 4.0 (TMm), while in Cornicabra oils obtained by extraction without (C) and with US application (usC), the pungent notes had an intensity of 3.25 and 3.75, respectively (Table 1). Except for TCm and usTMm, the pungent intensity was similar to that of thyme flavor. Moreover, in most of the situations, as observed for the thyme flavor, a decrease in pungency with USAE was observed. Previous results with the Galega cultivar showed that the thyme flavor detected by panelists was more intense in the trials performed with thyme addition in the mill (≥8) than in the malaxator (5–6), and the bitter taste had similar intensity in both fresh flavored oils [6]. The use of a different olive cultivar may explain these differences.

### 2.3. Storage Studies of Flavored Oil Preparations

#### 2.3.1. Fatty Acid and Phenolic Compositions

Oil flavored preparations should be characterized not only after the extraction process but also during storage. In fact, we want to obtain a product, which will preserve, over the storage period, the flavoring and bioactive compounds, which were eventually transferred from the thymes to the oil as well as to evaluate whether important changes in the chemical composition of the oils occur. Fatty acids are the main components of olive oil and are related to the nutritional value and to its resistance to oxidation [26]. Moreover, thymes are rich in essential oils, which, apart from the odorants transferred to the oil, can promote the increase/decrease in some fatty acids already present in the virgin olive oil [27]. To search for changes in fatty acid composition, the evaluation of all samples by NIR spectroscopy was performed at three time points (after extraction and at four and eight months of storage). The main fatty acid composition of the Cornicabra virgin olive oil and of oil flavored preparations is presented in Table 3. The Cornicabra oils have low contents of saturated fatty acids (13%) and high contents of polyunsaturated fatty acids (linoleic acid: 15.9%). The fatty acid composition of these Cornicabra oils is very different from the ones of the same cultivar produced in Spain, which have lower contents of PUFA (<4.5%) and higher oleic acid content (~80%) [28,29], but this is in accordance with previous results from Peres et al. 2019 [30] in Portugal.

Small changes in fatty acid composition seem to occur in olive preparations with mastic thyme (TM) for oleic, linoleic and palmitic acids, while for lemon thyme (TC), the amounts of these fatty acids are similar to those in the Cornicabra VOO. The samples from trials with US treatment showed similar fatty acid composition to those produced without US treatment.

The hydrophilic phenolic compounds of VOO belong to different classes: phenolic acids and derivatives, phenolic alcohols, lignans, secoiridoids and flavones [31]. The main antioxidant properties come from *o*-diphenols and are related to hydrogen donation, i.e., their ability to improve radical stability by forming an intramolecular hydrogen bond between the free hydrogens of their hydroxyl group and their phenoxyl radical [32]. Apart from antioxidant activity, hydroxytyrosol (Htyr) and its derivatives (e.g., oleuropein complex and tyrosol, Tyr) are phenolic compounds found exclusively in olive oil with “protective effect against oxidative stress on blood lipids”. Since 2012, the European Food Safety Authority (EFSA) allows the use of this health claim on labeling [33]. Thus, studying the presence of these compounds in olive oil preparations is of utmost importance. In the present study, we followed the behavior of total phenols over a storage period (after four and eight months of storage) of VOO and flavored preparations with mastic thyme or lemon thyme (Figure 1). US application, either in the VOO control or in flavored preparations with mastic thyme or lemon thyme, did not significantly improve the total phenolic content of the oils (Figure 1). During storage, a decrease in the phenolic content was observed for all samples. After four months of storage, the mastic thyme flavored coextracted preparations showed lower phenolic contents than the control samples. However, after eight months of storage, flavored oils were richer in phenolics than the controls, except the sample with thyme addition in the mill, in the absence of USAE (TMm). Moreover, the trials with TM (in the mill or malaxation) and US treatment showed lower decrease in total phenols.

After four months of storage, the phenolic profile of the oils was checked for the main phenolic components. Cornicabra oils showed the presence of the main phenolic compounds referred by the International Olive Council (IOC, 2007) (Figure 2). Thus, 3,4-DHPEA-EDA (oleacein or dialdehydic form of elenolic acid linked to hydroxytyrosol) was the main compound. Hydroxytyrosol (Hyt), tyrosol, vanillic acid, *p*-coumaric acid, cinnamic acid and vanillin, oleuropein, *p*-HPEA-EDA, pinoresinol, luteolin and apigenin were identified with authenticated standards. The presence of a high content of oleacein shows that the Cornicabra oils were extracted from fruits, which were still in good condition (i.e., healthy fruits not subjected to frost) [6]. Moreover, no improvement of phenolic compounds with USAE was observed (chromatogram overlap of lines).

TM enrichment promoted an improvement of several new phenolic compounds, especially when thyme was added in the malaxation step (TMb) (Figure 2). With TC addition, a decrease in several phenolic compounds was observed, namely oleacein (3,4-DHPEA-EDA) (peak 12), which showed a decrease of 80% in area, and fewer new phenolic compounds were identified. 

The main phenolic compound identified in *T. mastichina* is rosmarinic acid [34,35] (RT = 31.67 min). However, in all the flavored oil preparations, this phenol is almost absent or is overlapped by the peaks of oleuropein derivatives (Figure 2). This was also observed in a previous study [6], showing that this compound has a low solubility in oil. In fact, from the point of view of olive technology, the addition of water in the malaxation step has a high influence on the presence of phenolic compounds in the oil [36]. Previous trials with TM coextraction showed the importance of water addition in phenolic extraction [6]. Moreover, the proportion of phenolic compounds in the oil, water and pomace depends on the relative polarity of these compounds, as well as the presence of surfactants, the temperature used in VOO extraction, the composition and relative amounts of the phases [37]. For instance, 3,4-DHPEA-EDA (kp = 1.49) and 3,4-DHPEA-EA have a considerable solubility in olive oil, as opposed to oleuropein and Hyt (kp = 0.01) [37]. Therefore, with regard to thyme phenolic compounds, only caffeic acid—which is also slightly soluble in oil (kp = 0.09) [37]—vanillin, *p*-coumaric acid, o-coumaric acid, apigenin and peak 14, 19, 20 seemed to improve when TM was added. However, for vanillin, *p*-coumaric acid, o-coumaric acid, identification using the retention time of the standards is not conclusive because other compounds from herbs or from enzymatic or oxidation reactions, which may be formed during the extraction process, may elute at the same time. Peres et al. 2021 [6] reported an increase in two unidentified peaks (retention time of 40 and 50 min) in the coextraction of Galega olives with TM. 

Malaxation trials with *T. vulgaris* mainly introduced caffeic acid (17.6–29.8 g/kg), carnosol (12.4–22.3 mg/kg), protocatechuic acid (9.9–11.6 mg/kg) and rosmarinic acid (6.0–7.6 mg/kg) in the virgin olive oi. This increase was attributed to the presence of vegetation water in the olive paste, which can act as a solvent for the improvement of the extraction of herbs’ polar compounds [21]. In our case, the content of vegetation water was low (<45%), which can explain the low extraction of phenolic compounds from the thyme to the oil. This shows the importance of water in the malaxation step.

#### 2.3.2. Chlorophyll Pigments 

The coextraction with mastic thyme or lemon thyme results in flavored oil preparations with a greener color compared with the extra virgin olive oil. Thus, the total content of chlorophyll pigments was assayed in all samples (Figure 3) immediately after extraction and after 4 and 8 months of storage at room temperature and in the dark.

A significant improvement in chlorophyll pigments was observed in all coextraction trials, and the extraction of those pigments was higher with TC, particularly without US treatment (Figure 3). USAE had no significant impact on pigment removal during VOO extraction or the coextraction with *Thymus mastichina* (TM). When *Thymus citriodorus* (TC) was added in the mill, a significant decrease in pigment content was obtained (usTCm) compared to the coextraction without US pretreatment (TCm). During storage, a pigment decrease was mainly observed in the lemon thyme (TC) flavored preparations. 

#### 2.3.3. Multivariate Data Analysis of Flavored Oil Preparations after Storage

After 8 months of storage, all samples were analyzed in terms of quality parameters, phenolic and chlorophyll contents and main fatty acid composition. No significant differences were observed in fatty acid composition after storage. Regarding the other parameters, a quality decrease was observed, which was accompanied by a reduction in phenolics and pigment contents. The maximum acidity reached was 0.22%; the PV was 11.16 meq O_2_ kg^−1^; K_232_ reached a maximum of 2.42 and K_270_—0.24.

The phenolic content decreased more than 80% in VOO (from 294.7 to 50.5 mg GAE/kg VOO, average values), around 59% in the flavored preparations with TM (from 270.0 to 110.5 g GAE/kg oil, average values) and ca. 76% in the flavored preparations with TC (from 177.6 to 43.4 mg GAE/kg, average values) after 8 months of storage. The content of chlorophyll pigments was, on average, 37.6 mg pheophytin/kg in oils coprocessed with TM and 50.1 mg pheophytin/kg in oils coprocessed with TC. On average, immediately after extraction, the pigments in TM and TC preparations were 14- and 19-fold the values in the EVOO (2.62 mg pheophytin/kg). After 8 months of storage, a 35% decrease in pigments was observed in the VOO, while in oils flavored with TM or TC, a 4.5 and 15.7% decrease was observed, respectively. 

All chemical quality parameters, together with the phenolic and green pigment contents (total of six variables), were used to characterize all the oil samples along the storage period. Principal component analysis (PCA) combined with hierarchical cluster analysis (HCA) were used as an attempt to find relationships among variables and to detect eventual sample groups, i.e., to find a pattern in the whole dataset. PCA showed that the original six-dimensional space, defined by the initial variables, could be reduced to a plane defined by the first two new axes (principal components or factors). Both factors have eigenvalues bigger than 1.0 and are therefore significant. This plane explains more than 85% of the information contained in the original dataset (Figure 4a). In this case, the first axis (Factor 1) might be identified as the storage time axis: the values of PV, K_232_ and K_270_ increase along the positive side, indicating a quality decrease, which also appears to be related with the illustrative variable “time”. Along the negative side of this axis, the phenolic content increases, indicating that the biologic and antioxidant values of the samples increase in this sense. We have an increase in the chlorophyll pigments along the positive side of the second axis, while the acidity in-creases along its negative side. 

When all 30 oil samples are projected onto this plane (Figure 4b), we can see that the samples are distributed along the principal component 1 as a function of their age: the initial samples (EVOO and flavored oil preparations), as well as EVOO after 4 months of storage, are plotted on the second and third quadrants; the samples at the end of the storage are presented in quadrants 1 and 4, while the flavored oil preparations at 4 months of storage are plotted in the center of the plane. This means that the quality of VOO and flavored preparations decreases with time. Flavored preparations (TM or TC) are mainly placed in quadrants 1 and 2, indicating that they are well correlated with pigment content. Pigment degradation, mainly occurring in TM oils and VOO after 8 months of storage, is well illustrated in Figure 4b: these samples are located in quadrant 4, which indicates a lower pigment content, higher acidity, PV, K_232_ and K_270_ values. 

In the PCA plot, we see the projections of the samples placed in a six-dimension space onto a plane. To verify the presence of eventual sample groups, HCA was performed. The dendrogram is presented in Figure 5. At a single linkage distance of around 35, we can confirm the presence of five groups indicated in Figure 4b and the grouping trend suggested by PCA. The preparations coextracted with TM, at time zero, proved to be more similar to the control VOO at time zero and after 4 months of storage than the initial flavored preparations with lemon thyme (TC). The flavored samples are mainly grouped by age. Thyme addition in the mill or in the malaxator and the use of USAE proved not to be the main factors discriminating flavored oil preparations.

#### 2.3.4. Oxidative Stability 

The oxidative stability (OS) of oil preparations obtained by coextraction with mastic thyme or lemon thyme was evaluated just after extraction (without storage) and after 8 months of storage at 23 °C. Figure 6 shows that after extraction, all the oils with thyme addition presented a lower OS compared with the controls (C and usC), without significant differences with USAE application. The samples coextracted with mastic thyme (TM) showed higher OS than the oils coextracted with TC, corresponding to their higher phenolic compound contents. After 8 months of storage, a decrease of 41% in OS was achieved in the control trial sample, while with TM, a 24% (TMb) and 15% (usTMm) decrease was found; for TC samples, no significant differences in OS were observed after storage.

## 3. Discussion 

The possibility of converting long malaxation times—which result in the loss of bioactive compounds—into a complete thermal and ultrasonic exchange, which requires only a few minutes, is a goal for olive oil technology [38]. However, the types of ultrasonic systems (probe or bath) and the operating conditions of frequency, process time and the type of food matrix influence the acoustic cavitation performance [39].

The possibility of preparing new flavored olive oils by USAE, without damaging the final oil, was tested in the present work. The resulting oils showed an improvement of thyme flavor, and no changes in chemical quality parameters were observed. Moreover, USAE did not affect the extraction of phenolics and chlorophyll pigments from the thymes to the oils. This may be explained by the use of overripe olives with a low moisture content. In fact, it was observed that the effect of USAE is progressively reduced with olives at a higher ripening stage [14]. Additionally, previous results by Clodoveo et al. 2013 [16] demonstrated that the thermal effect of ultrasounds led to a quick heating of olive paste, thus reducing the malaxation time, independently of the different technological performance of the varieties. 

In all our trials, the temperature achieved in the paste was between 24 and 29 °C, which is the temperature range indicated for obtaining high-quality EVOO. Moreover, the ultrasound treatment did not change the quality indices of VOO, such as free acidity, the peroxide value and K_232_ and K_270_. In addition, total phenols were not significantly affected by USAE. On the contrary, Clodoveo et al. (2013) [16] found a decrease in phenolic compounds obtained with USAE, mainly explained by the action of several enzymes (i.e., beta-glucosidase, peroxidases, polyphenol oxidases), which are activated by US in the presence of oxygen. These enzymes are associated with oxidative catabolism, acting both on glycosides and on the derived hydrophilic phenols [40].

Our results show that the impact of USAE in the coextraction trials performed with thyme addition was not relevant under the conditions used.

The present work shows that the enrichment of olive oils with phenolic compounds or pigments by coextraction is very dependent on the thyme used. In fact, the enrichment with phenolic compounds is much more significant with TM than with TC. Oils coextracted with TC presented lower total phenolic contents than the original EVOO or oils coextracted with TM. Oils coextracted with TC showed an important decrease in specific phenolics of olives/olive oil, and the enrichment with specific phenols from this thyme was very low. The samples obtained with TM coextraction showed chromatograms with new phenolic compounds, which have a thyme origin. Moreover, the addition of thyme in the malaxation step proved to be better than the addition during milling in terms of the improvement of phenolic compounds in the oils. However, previous coextraction studies with a different cultivar (“Galega vulgar”) showed better results for phenolic compounds when TM was added in the mill, which may be explained by use of a TM of a different origin [6].

Concerning pigment extraction, the enriched oil samples were characterized by a green color due to a high content of chlorophyll pigments, especially when TC was used. The presence of higher amounts of green pigments with pro-oxidant activity [41] in TC flavored oils, together with a lower phenolic content, might explain the lower oxidative stability of these preparations. 

The coextraction of Cornicabra olives with TM produced flavored oils richer in phenolic compounds and more resistant to oxidation than those obtained by coextraction with TC. The main advantage of these coprocessed oils is the presence of the thyme flavor in the oils extracted from ripe olives with olive fruity flavor notes, bitterness and pungency at low intensity. These new products may be an option for increasing the value of the original VOO obtained from overripe olives. 

## 4. Materials and Methods

### 4.1. Materials

Olive fruits of the “Cornicabra” cultivar used in the present study were produced in a rain-fed olive grove situated in Figueira de Castelo Rodrigo, Beira Alta Region, Portugal, and were kindly offered by the producer Rui Torres. The fruits were picked in January 2022, at the end of the harvest season, with a ripening index (RI) of 6.0, with a very low water content (43.93% ± 0.11). They were used for the coprocessing experiments with thyme addition in the crushing or malaxation operation with US application. Dried *Thymus mastichina* L. (TM) and *Thymus citriodorus* (TC) (lemon thyme), with moisture content of 11.5 and 10.5% (wet basis), respectively, were purchased from Ervas de Zoé, Ladoeiro, Portugal, and were produced according to organic farming (OF) guidelines. The dried plants were subjected to milling in a home-mill (Vorwerk Thermomix TM31, Vorwek International Mittelsten Scheid & Co., Ltd., Wollerau, Switzerland). Sieve analysis was performed to classify the particles according to their size using five sieves of the Tyler equivalent series (10, 28, 35, 60 and 140 mesh, equivalent to opening sieves of 1.68, 1, 0.42, 0.25, 0.106 mm, respectively): more than 90% of thyme granulometry was [1;1.68 mm[(93% TC; 96% TM)]].

### 4.2. Methods

#### 4.2.1. Coextraction and US Assisted Trials

The olives were crushed with a hammer mill equipped with a 4 mm sieve at 3000 rpm. To evaluate the effect of ultrasound (US) treatment on polyphenol extraction, the olive paste was subjected to ultrasounds in a Bandelin Sonorex Digitec 10P bath (Bandelin Electronic, Berlin, Germany), with a frequency of 35 kHz, power 10 for 10 min, followed by a 20 min malaxation (27–30 °C) in a laboratory oil extraction system (Abencor analyser; MC2 Ingenieria y Sistemas S.L., Seville, Spain). Thyme addition (TM or TC) was performed in the mill (m) or the malaxator (b) at a dose of 2.5% (m/m). The experiment was carried out in triplicate, and the results were compared with those of the control (30 min malaxation without US treatment). 

After malaxation, the olive oil was recovered by centrifugation at 3500 rpm for 60 s and analyzed for its total phenolic content. For each batch, 4 independent extractions were performed. The results were handled using the Software Statistica, version 7, from StatSoft, Tulsa, USA.

#### 4.2.2. Characterization of Olive Oils

The acidity (% FFA (expressed as oleic acid)), peroxide value (PV) (meq O_2_ kg^−1^), UV absorbances (K_232_ and K_270_) and the major fatty acids (oleic, linoleic, palmitic and stearic) were evaluated by NIR spectroscopy (MPA, Bruker Optics, Ettlingen, Germany). The calibration model B-Olive-Oil (Bruker Optics, Ettlingen, Germany) was used in accordance with a previously described method [6]. Fatty acid complete composition of Cornicabra oil (Control) was checked through gas chromatography and evaluated as fatty acid methyl esters by GC-FID in a Hewlett Packard 6890 (SP column 2380^TM^ Supelco (60 m × 0.25 mm × 0.20 μm)) [42].

A Metrohm Rancimat model 670 (Herisau, Switzerland) was used to evaluate oxidative stability (OS) using the following conditions: temperature of 120 °C; air flow of 20 L h^−1^. Chlorophyll pigments were assayed using the method proposed by Pokorný et al. (1995) [43]. All samples (15 mL of each oil in a specially made blue glass covered with a watch glass to concentrate the volatile compounds) were also sensory evaluated by a trained panel with a profile sheet, where an unstructured 10 cm length scale was used to mark the intensity of the descriptors (mainly thyme/lemon flavor, bitter and pungent intensities and eventual defects), as previously described [6].

For total phenols’ evaluation, 500 mg of olive oil was extracted with 1 mL of a methanol/water mixture (80:20, *v*/*v*) in 2 mL Eppendorf reaction tubes. After vigorous shaking for 1 min using a vortex, the sample was centrifuged (Eppendorf MiniSpin Plus Microcentrifuges, Eppendorf, Madrid) at 13,400 rpm for 5 min at 20 °C. This process was performed 3 times. The 3 extracts were combined and the volume adjusted to 5 mL with ultrapure water. The quantitative determination of phenolic content is based on the reaction of the Folin–Ciocalteau reagent with the functional hydroxy groups of phenolic compounds. In a cuvette of 1 cm in width, for spectrophotometric analysis, 0.1 mL of the aqueous-methanolic solution of phenolic compounds extracted from the VOO was diluted in 1.5 mL of ultrapure water, followed by the addition of 0.1 mL of the Folin–Ciocalteau reagent, and maintained for 3 min. Then, 0.3 mL of 20% (*w*/*v*) sodium carbonate aqueous solution was added and mixed. The absorbance of the solution was measured after 1 h against a blank sample using a UV–VIS spectrophotometer at a wavelength of 765 nm (JASCO 7800, Tokyo, Japan). The calibration curve was constructed using standard solutions of gallic acid (R^2^ > 0.999). The results were expressed as milligram of gallic acid per kilogram of oil (mg GAE/kg). 

The phenolic compounds’ profile was evaluated by HPLC. An Agilent 1100 HPLC system (Agilent, Santa Clara, CA, USA), consisting of a degasser, a quaternary pump, a column oven, an autosampler and a UV detector, was used. The stationary phase consisted of a Purospher C18 analytical column (250 mm × 3.9 mm × 4 µm). The mobile phase consisted of solutions of (A) 0.2% H_3_PO_4_ (*v*/*v*), (B) methanol and acetonitrile at a constant flow rate of 1 mL min^−1^. The gradient program used was the one indicated by the IOC document [44]. Sample preparation was performed according to Pirisi et al. 2000 [45]: 2 g of oil was weighed in a centrifuge tube and added to 1.0 mL of *n*-hexane and 2.0 mL of methanol water 80:20 (*v*/*v*). The mixture was stirred for 2 min in a vortex apparatus, and the tube was centrifuged at 3000 rpm for 3 min (Ortoalresa, mod Lince, Madrid, Spain). The methanol layer was separated, and the extraction repeated twice. The extracts were combined and washed twice with 2 mL of n-hexane. The *n*-hexane was discarded, and the methanolic solutions were evaporated (Buchi Rotavapor R-114, Büchi, Flawil, Switzerland) to dryness under reduced pressure and low temperature (<35 °C). The residue was dissolved in 500 µL of methanol solution. Before injection, filtration with the Pall Gelman Acrodisc (membrane 0.45 μm, 25 mm, GHP) of the sample was performed.

Standards of tyrosol, vanillic acid, vanillin, caffeic acid, ferulic acid, *o*-coumaric acid, *p*-coumaric acid, oleocanthal, apigenin, rosmarinic acid, and quercetin were purchased from Merck (Darmstadt, Germany); thymol, hydroxytyrosol, oleuropein, luteolin from Extrasynthese (Genay, France); carvacrol, myricetin, pinoresinol and taxifolin from TCI Europe (Zwijndrecht, Belgium); and oleacein from TRC (Toronto, ON, Canada). They were used for the identification of phenolic compounds.

#### 4.2.3. Statistical Analysis 

To evaluate whether significant differences existed among samples concerning the quality criteria, phenolic profile and pigments during oil storage, one-way ANOVA (post hoc Sheffé test was used; *p* ≤ 0.05) was performed. Multivariate data analyses, namely principal component analysis (PCA) and hierarchical cluster analysis (HCA), were carried out on a matrix consisting of 30 rows (30 samples) and 6 columns, corresponding to the following active variables: free fatty acids (FFA), PV, K_232_, K_270_, total phenolic compounds (“Phenols”) and total chlorophyl pigments (“Pigments”). Storage time was used as an illustrative or supplementary variable (“Time”), which means that this variable was not used to build principal components, but it might have helped in interpreting the dimension of the variability of the data. Principal components (new axis) with eigenvalues higher than one were considered as significant and were therefore retained in the analysis [46,47]. For HCA, the samples were grouped based on their average Euclidean distance, while the single linkage method was used for sample aggregation [46,47]. ANOVA and multivariate data analysis were performed using the software Statistica, version 7, from Statsoft, Tulsa, OK, USA.

## 5. Conclusions

Cornicabra ripe olives (ripening index of 6.0) with olive fruity flavor notes, bitterness and pungency at low intensity, showed to be adequate for the coextraction with mastic thyme (TM) or lemon thyme (TC), added either during the milling or the malaxation steps. The coextracted oils exhibited a pleasant thyme flavor, without changing their chemical quality. The coextraction of Cornicabra olives with TM produced flavored oils richer in phenolic compounds and more resistant to oxidation than those obtained by coextraction with TC. Moreover, higher content of phenolic compounds in the oils was observed when thyme was added in the malaxation step, instead of during the milling operation. The flavored oils were also characterized by a green color due to a high content of chlorophyll pigments, extracted from the thyme to the oil during coextraction, especially when TC was used.

The ultrasound assisted extraction, carried out in some coextraction trials before the malaxation operation, did not affect the quality parameters or the contents of total phenols and green pigments in the flavored oils. Therefore, when Cornicabra ripe olives with a low moisture content are used, the USAE is not needed.

The production of these new flavored oils by coextraction with thyme may be an option for increasing the value of the original VOO obtained from overripe olives.

## Figures and Tables

**Figure 1 molecules-28-06898-f001:**
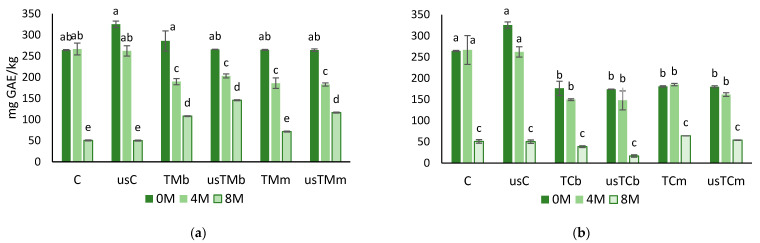
Total phenols after storage (0M—without storage; 4M and8M—after 4 and 8 months of storage) for the virgin olive oil (control) and the oils obtained in the trials with mastic thyme (**a**) and lemon thyme (**b**), either submitted to US treatment or not (C—control; us—trials with USAE; TM—*Thymus mastichina*; TC—*Thyme citriodorus*; b—beating; m—mill) (for each storage period and each thyme, different letters mean differences between trials at *p* < 0.005; Sheffé test).

**Figure 2 molecules-28-06898-f002:**
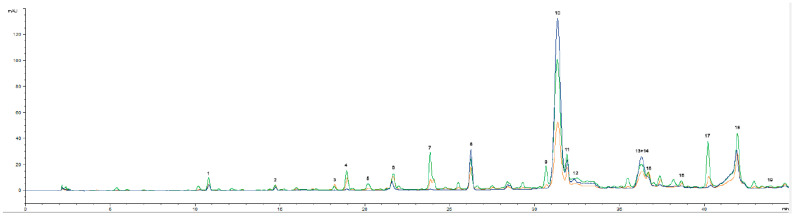
Phenolic profile of Cornicabra olive oils (C—blue), *T. mastichina* flavored olive oil preparations (TMb—green; TMm—orange) at 280 nm (1—hydroxytyrosol; 2—tyrosol; 3—vanilic acid; 4—caffeic acid; 5—vanillin; 6—*p*-coumaric acid; 7—unidentified; 8—unidentified; 9—*o*-coumaric acid; 10—3,4-DHPEA-EDA; 11—oleuropein; 12—rosmarinic acid; 13 + 14—oleocanthal + pinoresinol; 15—cinnamic acid; 16—luteolin; 17—unidentified; 18—unidentified; 19—apigenin).

**Figure 3 molecules-28-06898-f003:**
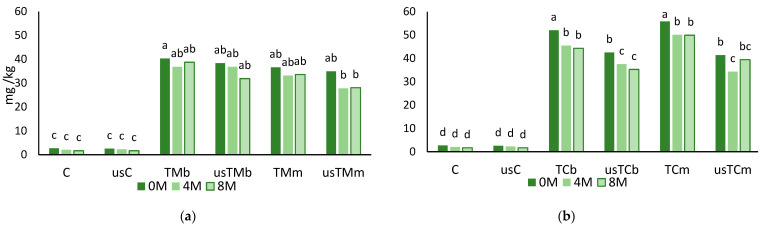
Chlorophyll pigments (mg pheophytin/kg) after storage (0M—without storage; 4M and 8M—after 4 and 8 months of storage) for virgin olive oil and oils obtained in the trials with mastic thyme (TM) (**a**) and lemon thyme (TC) (**b**), either submitted to US treatment or not (C—control; us—trials with USAE; TM—*Thymus mastichina*; TC—*Thymus citriodorus*; b—beating; m—mill) (for each storage period and each thyme; different letters mean differences between trials at *p* < 0.005; Sheffé test).

**Figure 4 molecules-28-06898-f004:**
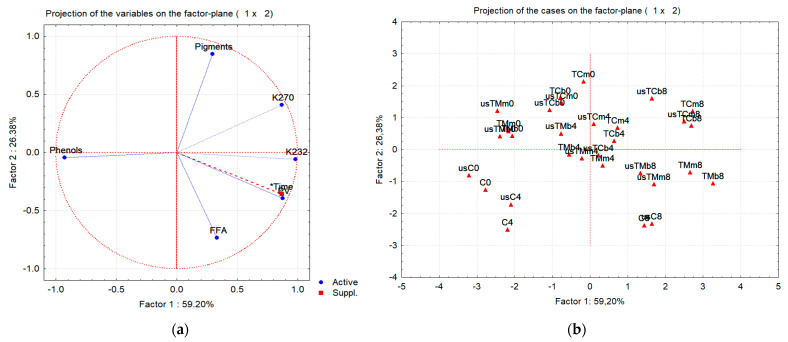
PCA of all oil samples characterized by chemical quality parameters, phenolic and chlorophyll pigment contents after storage (0—without storage; 4 and 8—after 4 and 8 months of storage) for virgin olive oil (samples C) and oils obtained in the trials with mastic thyme (TM) and lemon thyme (TC), either submitted to US treatment or not (us—trials with USAE; b—beating; m—mill). (**a**) Plot of the variable loadings on the plane defined by Factor 1 and Factor 2 (*Time- storage time was added as illustrative/supplementary variable); (**b**) Plot of the oil samples on the plane defined by Factor 1 and Factor 2.

**Figure 5 molecules-28-06898-f005:**
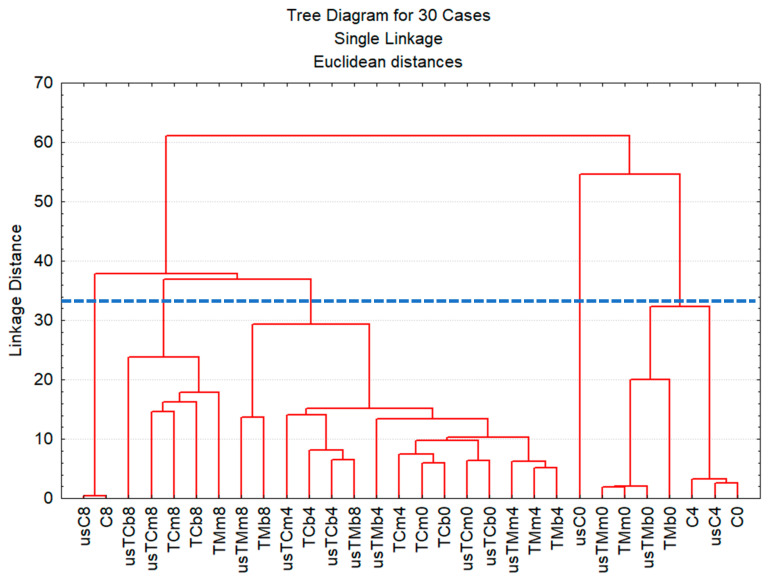
Dendrogram of all oil samples characterized by chemical quality parameters, phenolic and chlorophyll pigment contents after storage (0—without storage; 4 and 8—after 4 and 8 months of storage) for virgin olive oil (samples C) and oils obtained in the trials with mastic thyme (TM) and lemon thyme (TC), either submitted to US treatment or not (us—trials with USAE; b—beating; m—mill).

**Figure 6 molecules-28-06898-f006:**
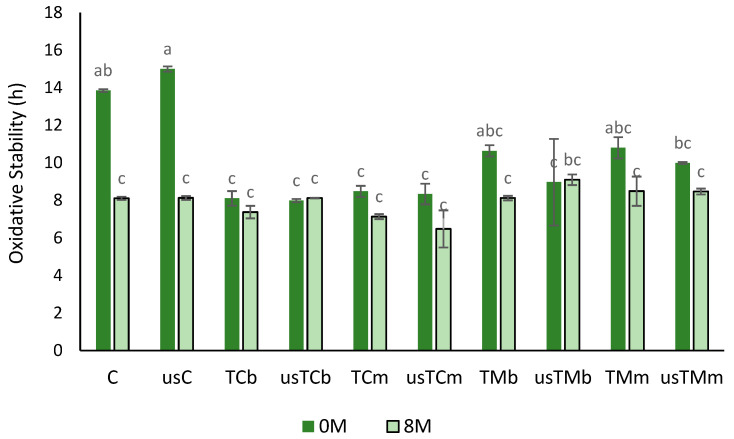
Oxidative stability (hour) after storage (0M—without storage; 8M—after 8 months of storage) for VOO (control) and oils obtained in the trials with mastic thyme and lemon thyme, either submitted to US treatment or not (C—control; us—trials with USAE; TM—*Thymus mastichina*; TC—*Thymus citriodorus;* b—beating; m—mill) (different letters mean differences between trials at *p* < 0.005; Sheffé test).

**Table 1 molecules-28-06898-t001:** Quality criteria of Cornicabra oils after extraction without (C) and with US application (usC).

Quality Criteria	Olive Oils	
	C	usC
Acidity (% oleic acid)	0.17 ± 0.00	0.16 ± 0.01
Peroxide value (meq O_2_ kg^−1^)	5.35 ± 0.39	5.39 ± 0.28
K_270_	0.14 ± 0.01	0.15 ± 0.01
K_232_	2.01 ± 0.00	1.97 ± 0.00
Median of defect	0	0
Olive ripe fruity	1.95 ± 0.21	2.2 ± 0.14
Bitter	1.50 ± 0.71	2.0 ± 0.71
Pungent	3.25 ± 0.35	3.75 ± 0.36

**Table 2 molecules-28-06898-t002:** Quality criteria of flavored oil preparations (olives and lemon thyme, TC, or mastic thyme, TM) added in the mill (m) or in the thermobeater (b), obtained without any ultrasound treatment or with ultrasound (us) application to the paste before malaxation, and before storage at 23 °C (0M).

Quality Criteria	TCb	usTCb	TCm	usTCm	TMb	usTMb	TMm	usTMm
Acidity (% oleic acid)	0.16	0.16	0.14	0.15	0.17	0.17	0.16	0.14
Peroxide value (meq O_2_ kg^−1^)	4.68	4.75	6.17	5.54	4.97	4.47	4.79	4.97
K_270_	0.18	0.17	0.20	0.19	0.16	0.14	0.16	0.16
K_232_	2.14	2.12	2.16	2.14	2.04	2.03	2.03	1.98
Median of defect	0	0	0	0	0	0	0	0
Thyme flavor	3.5	1.5	3.0	2.5	4.7	3.75	4.9	4.45
Bitter	1.0	0.85	1.25	1.5	0.7	1.05	1.6	1.5
Pungent	3.25	2.25	1.75	2.65	3.6	2.6	4.0	1.2

**Table 3 molecules-28-06898-t003:** Fatty acid composition (major fatty acids) of Cornicabra virgin olive oil (C) and of oil preparations flavored with thyme (with and without USAE). The values are the average values of all evaluations performed at 0, 4 and 8 months of storage (the meaning of samples is presented in the heading of Table 2).

Fatty Acids	C	TCb	usTCb	TCm	usTCm	TMb	usTMb	TMm	usTMm
Oleic acid (C18:1)	68.3	68.2	68.3	68.2	68.4	69.3	69.1	69.3	69.3
Linoleic acid (C18:2)	15.9	15.9	15.9	15.8	15.8	15.1	15.0	15.1	14.9
Palmitic acid (C16:0)	9.7	9.5	9.5	9.6	9.6	10.1	10.1	10.0	10.1

## Data Availability

Data are available from the authors upon request.

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
