# Peer review of "Ultrasound Assisted Coextraction of Cornicabra Olives and Thyme to Obtain Flavored Olive Oils"

_molecules, 2023, doi:10.3390/molecules28196898_

Round 1
Reviewer 1 Report
The comments are as follows:
1. The authors should include the overall conclusion in the abstract.
2. Introduction - What is the novelty of this work in comparison to similar ones reported previously?
3. Please, include the moisture content of the dried sample material.
4. What was the temperature during UAE?
5. More recent studies should be cited.
Author Response
Dear Colleague,
First, we would like to thank you for taking the time to review our manuscript submitted to Molecules and for your comments that helped us to improve its quality.
For your convenience, we have included in the attached file, the itemized list or your comments and our replies according to changes and modifications that have now been made to the original manuscript (highlighted in yellow).
We hope that you will find our manuscript improved, and that you now feel it is of sufficient quality for publication in this Special Issue.
Best regards,
Suzana Ferreira-Dias

Reviewer 2 Report
The manuscript is well composed and prepared. Plese find the following small remarks helping to improve it. Please provide the sample codes in Figure 5 (all figures should be self-reading). I feel a lack of description of sensory analysis methodology. Please unify the list of references, paying attention to the article titles formatting.
Author Response
Dear Colleague,
First, we would like to thank you for taking the time to review our manuscript submitted to Molecules and for your comments that helped us to improve its quality.
For your convenience, we have included below the itemized list or your comments and our replies according to changes and modifications that have now been made to the original manuscript (highlighted in yellow).
We hope that you will find our manuscript improved, and that you now feel it is of sufficient quality for publication in Molecules.
Best regards,
Suzana Ferreira-Dias

Reviewer 3 Report
1) Table 1 and Table 2 – abreviations PV, K270 and K232 must be explained in title of table or in notes under table.
2) Figure 1, Figure 2 and Figure 4: VOO must be writen fully not in abreviation in titles of figures.
3) Figure 4: – figure b) doesn't make sense in a work context and must be deleted.
4) Figure 5: grouping doesn't make sense in a work context. Also this Figure only repeats Figure 4b.
5) Why was not analysed oksidative stability after 4 months?
6) The second and the third chapters of manuscript (2. Results and Discussion and 3. Discussion, respectively) overlaps. The discussion part from chapter no. 2 must be moved to chapter no. 3.
Author Response
Dear Colleague,
We would like to thank you for taking the time to review our manuscript submitted to Molecules and for your evaluation.
For your convenience, we have included below the itemized list or your comments and our replies according to changes and modifications that have now been made to the original manuscript (highlighted in yellow).
We hope that you will find our manuscript improved, and that you now feel it is of sufficient quality for publication in Molecules.
Best regards,
Suzana Ferreira-Dias
Suzana Ferreira-Dias

Reviewer 4 Report
In this work, an ultrasound-assisted extraction method is presented to obtain thyme flavored oils from Cornicabra olives, having an aroma more appreciated by consumers. Mastic thyme and lemon thyme were used to flavor these oils. Regarding the production of olive oils enriched with complementary phenolic compounds derived from thyme, oregano and aromatic herbs, the literature is quite rich. Therefore I do not find the work of average originality. However, considering the importance of olive oil for its multiple recognized benefits on human health, I believe that any research in this field deserves attention, also considering that the aroma of the product is among the first qualities appreciated by consumers. The authors demonstrate solidly, with adequate presentation, that the process they propose allows obtaining parameters for the enriched oils sometimes better than the controls, although they find that the selected thyme species and the addition step are important for the enrichment of the polyphenolic fraction, for the extraction of chlorophyll pigments and shelf life.In my opinion the work is scientifically valid and adequately interesting for readers. I congratulate the authors for paper that I find scientifically valid and sufficiently interesting for readers.
Author Response
Dear Colleague,
We would like to thank you for taking the time to review our manuscript submitted to Molecules and for your positive evaluation.
Best regards,
Suzana Ferreira-Dias